# Eco-evo-devo implications and archaeobiological perspectives of trait covariance in fruits of wild and domesticated grapevines

**Vincent Bonhomme**[1,2☯]*, **Sandrine Picq**[1,2,3☯], **Sarah Ivorra**[1,2], **Allowen Evin**[1,2],
**Thierry Pastor**[1,2], **Roberto Bacilieri**[4], **Thierry Lacombe**[4], **Isabel Figueiral**[1,2], **Jean-Frédéric Terral**[1,2‡], **Laurent Bouby**[1,2‡]

**1** ISEM, Univ Montpellier, CNRS, EPHE, IRD, Montpellier, France, **2** Équipe « Dynamique de la biodiversité, anthropo-écologie », CC065 Montpellier Cedex 5, France, **3** Laurentian Forestry Centre, Natural Resources Canada, Québec, Canada, **4** UMR AGAP, Univ Montpellier, CIRAD, INRA, Montpellier SupAgro, Equipe « Diversité, Adaptation et Amélioration de la Vigne », Montpellier, France

☯ These authors contributed equally to this work.
‡ These authors also contributed equally to this work.
* bonhomme.vincent@gmail.com

**Data Availability Statement:** The full tables analysed are available (and citable) as: Bonhomme V, Picq S, Ivorra S, Pastor T. BerryPip dataset. doi:10.6084/m9.figshare.12696602.v2.

## Abstract

The phenotypic changes that occurred during the domestication and diversification of grapevine are well known, particularly changes in seed morphology, but the functional causes and consequences behind these variations are poorly understood. Wild and domesticate grapes differ, among others, in the form of their pips: wild grapes produce roundish pips with short stalks and cultivated varieties have more elongated pips with longer stalks. Such variations of form are of first importance for archaeobotany since the pip form is, most often, the only remaining information in archaeological settings. This study aims to enlighten archaeobotanical record and grapevine pip development by better understanding how size and shape (co)variates between pip and berry in both wild and domesticated *Vitis vinifera*. The covariation of berry size, number of seeds per berry ("piposity"), pip size and pip shape were explored on 49 grapevine accessions sampled among Euro-Mediterranean traditional cultivars and wild grapevines. We show that for wild grapevine, the higher the piposity, the bigger the berry and the more elongated the pip. For both wild and domesticated grapevine, the longer is the pip, the more it has a "domesticated" shape. Consequences for archaeobotanical studies are tested and discussed, and these covariations allowed the inference of berry dimensions from archaeological pips from a Southern France Roman site. This systematic exploration sheds light on new aspects of pip-berry relationship, in both size and shape, on grapevine eco-evo-devo changes during domestication, and invites to explore further the functional ecology of grapevine pip and berry and notably the impact of cultivation practices and human selection on grapevine morphology.

**Funding:** ANR project "Vignes et vins en France du Néolithique au Moyen Âge. Approche intégrée en archéosciences" (PI: Laurent Bouby) https://anr.fr/Projet-ANR-16-CE27-0013.

**Competing interests:** The authors have declared that no competing interests exist.

## Introduction

Grapevine (*Vitis vinifera* L.) is one of the most cultivated fruit species in the world [1], and has held a central economic and cultural role since ancient times, particularly in the Mediterranean area [2, 3]. The berries of grapevine are primarily used in wine production, but can be consumed fresh or dried (i.e. table grape). The wild progenitor of grapevine, *Vitis vinifera* subsp. *sylvestris*, was first domesticated in the South Caucasian area [4], which has yielded the oldest wine making evidence [5], dated to early Neolithic period (~8000 BP). The existence of other domestication centres has also been argued [6, 7]. Since the early times of domestication, grapevine varieties (or cultivars or "cépages") of *Vitis vinifera* subsp. *vinifera* have been selected and propagated; today there are several thousand varieties, identified by ampelography (i.e. grape morphology) and molecular markers [8, 9]. *V. vinifera* subspecies differ in their reproductive biology, and other phenotypic changes following domestication include larger bunches, larger berries, higher diversity in berry shape and skin colour, and higher sugar content [10, 11].

The quantitative morphological description of archaeobotanical material has brought major insights into the intertwined relationships between humans and domesticated plants [12–20], including grapevine [21–27]. So far, molecular approaches on ancient grapevine have yielded limited information on domestication [28, 29], with the study of ancient DNA hindered by its poor preservation in charred archaeobotanical material (but see [30]).

Wild and domesticate grape seeds differ in their form (size plus shape): wild grapes produce roundish pips with short stalks and cultivated varieties produce more elongated pips with longer stalks [31]. Such form variations have been identified on archaeological grapevine pips [22, 24, 25]. Archaeological material is often charred which can cause domesticated pips to appear more similar to wild pips [32], yet experimental charring has demonstrated the robustness of identification [26, 33].

The functional causes and consequences behind the form variation of grapevine pip are poorly understood. If size, shape, taste and colour of berries are traits that have been selected by humans, pip shape was likely not a direct target of selective pressures but may possibly be affected by: the berry size; the number of pips per berry; the growing environment and cultivation practices; the domestication status and the variety for domesticated grapevine; and developmental stochasticity. For instance, previous works suggested that pip size and the number of pips per berry are positively correlated to berry size [8, 34, 35] particularly for wild grapevines [24].

To what extent the form changes observed in archaeological pips imply changes in the form of berries? How this could be affected by the cultivation and domestication of wild individuals?

This paper scrutinizes how the form of berries and pips they contain covariate. A dataset of domesticated and wild contemporary grapevines allowed to compare patterns of covariation between wild and domestic *Vitis vinifera* compartments. This article is divided into four questions: i) how does size (co)vary between pips and berries, and depending on the number of pips?; ii) same question for shape; iii) how much pip shape depends on berry size, number of pips per berry, status, accession, and which practical consequences for archaeobotanical studies?; iv) can we infer the berry dimensions from archaeological pips?

## Results

### Preliminary analyses on modern material

The average piposity is equivalent between domesticated and wild accessions (mean±sd: domesticated = 2.01±0.891, wild = 2.1±0.968; GLM with Poisson error: df = 1468, z = 1.234,

P = 0.217 –Fig 1). The distribution of piposity, however, does differ (Fisher's exact test: P = 0.004) due to a higher proportion of 4-pips berries in wild grapevines (with 4-pips berries removed: P = 0.6272). No difference was observed between cultivated (domesticated and wild) accessions and those collected from wild (GLM with Poisson error: df = 1468, z = -0.676, P = 0.499).

## Covariation between pip and berry size in relation to the number of pips

**Wild vs. domesticated.** All berries and pips measurements were overall smaller for wild accessions compared to their domesticated counterparts (Wilcoxon one-tailed rank tests: all $P<10^{-8}$ –S1 Fig). Differences between wild and domesticated varied among the pip dimensions ($pip_{LengthStalk} > pip_{PositionChalaza} > pip_{Length} > pip_{Thickness} > pip_{Breadth}$).

Overall, the higher the piposity, the lower the contrast between domesticated and wild (Fig 2A). Pip dimensions of wild grapevines increase more substantially with piposity than their domesticated counterpart decrease. In wild grapevines, larger berries have more and larger pips (Fig 2A). No differences in berry dimensions/mass along increasing piposity were found for the cultivated grapevines, excepted between 1- and 2-pip for $pip_{Thickness}$ (Wilcoxon rank test: P = 0.006).

**Table vs. wine cultivars.** Table grapes have higher dimensions than wine varieties (Fig 2B). With increasing piposity, table varieties have bigger berries which is not the case for wine varieties. For pips, the only difference between low and high piposity were found for wine varieties and for $pip_{Breadth}$ and $pip_{Thickness}$ ($P<10^{-16}$).

**Wild grown in collection vs. wild in natura.** Wild grapevine pips and berries are bigger when in cultivation than their counterparts growing *in natura* (Fig 2C). Otherwise, trends of all measured variables are similar along increasing piposity. The berry mass ratios, relatively to wild collected *in natura*, were on average, 6.4 for wine varieties, 15.6 for table ones and 1.8 for cultivated wild.

Bivariate comparisons (S2 Fig) indicate positive correlations between all measurements. The total $pip_{Length}$ is the most consistent variable, between domesticated and wild grapevines: indeed, only the correlation with the $pip_{LengthStalk}$ show a significant interaction. Inversely, the correlations implying $pip_{LengthStalk}$ always show a significant interaction. For pips dimensions, the best correlations were found between $pip_{Length}$ and $pip_{PositionChalaza}$ (adj. $r^2 = 0.8$) among those with non-significant interactions, and between $pip_{LengthStalk}$ and $pip_{PositionChalaza}$ (adj. $r^2_{wild} = 0.615$, adj. $r^2_{domesticated} = 0.717$) among those with significant interactions. Compared to pips dimensions, correlations between berry dimensions were much better and the three possible interactions were all significant.

## Covariation between pip and berry shape in relation to the number of pips

The PCA shows that the first two PCs (Fig 3) gathered 69% of the total shape variation, and higher rank components levelled off (PC1 = 43.0%; PC2 = 26.2%; PC3 = 6.7%; PC4 = 3.9%), only the first two PCs were used as synthetic shape variables. Shape differences between wild and domesticated grapevines are mostly captured on PC1 yet scores on both PC1 (Wilcoxon rank tests, $P<10^{-16}$) and on PC2 ($P<10^{-16}$) were found different. PC1 represents how prominent is the stalk and how round is the pip; PC2 represents the circularity, a more global length/width ratio of pips, for the two views.

Regarding shape versus pip dimensions, $pip_{Length}$, correlated to all other measurements, is itself correlated with position on PC1. Two regressions were justified (Analysis of covariance: df = 1, F = 362.7, $P<10^{-16}$); their slope were identical (df = 1, F = 0.037, P = 0.848) but their intercept differed between wild and domesticated. These two regressions were significant yet

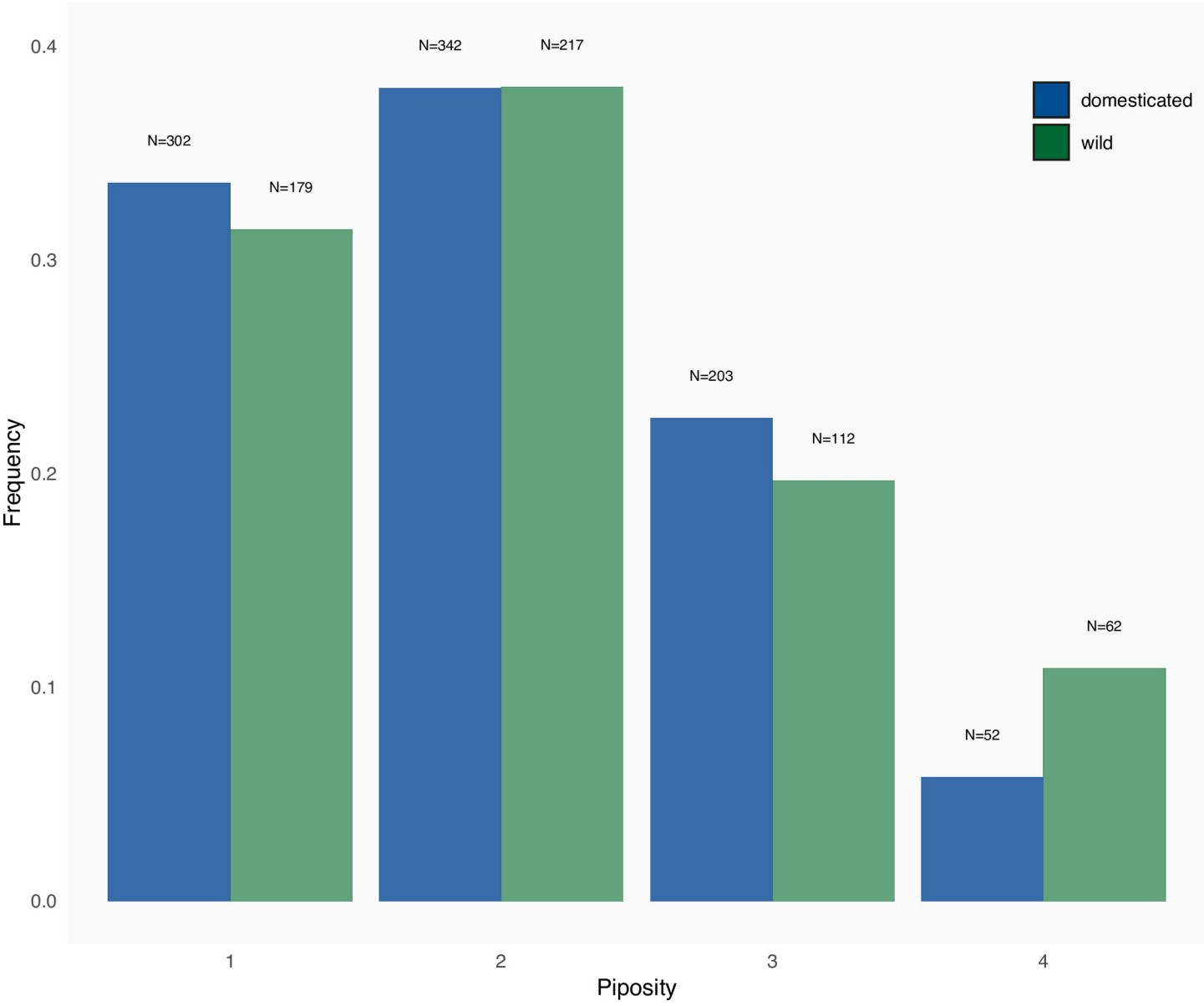

**Fig 1. Distribution of the number of pips per berry for wild and domesticated grapevines.**

$r^2$ were low (wild: $P<10^{-16}$, adj. $r^2 = 0.195$; domesticated: $P<10^{-16}$, adj. $r^2 = 0.240$ –Fig 4A). When PC1 and PC2 are considered jointly, two regressions were not justified (P = 0.04) and the $r^2$ was lower (P = 0.04, adj. $r^2 = 0.181$ –Fig 4A). The longer the pip is, the more "domesticate" it looks, particularly in terms of stalk prominence.

As concerns shape versus piposity, the latter is associated with shape changes on PC1 between wild and domesticated both overall (see above) and within levels (Wilcoxon rank tests, all $P<10-10$ –Fig 4B). Within domesticated accessions, differences were never significant. Within wild accessions, differences were not found between pairs of successive piposity levels but those between 1–3, 2–3 and 2–4 (all with $P<10-8$ –not shown). For PC2, differences observed between domesticated and wild vanished for high piposity (1-pip: $P<10^{-12}$; 2-pips:

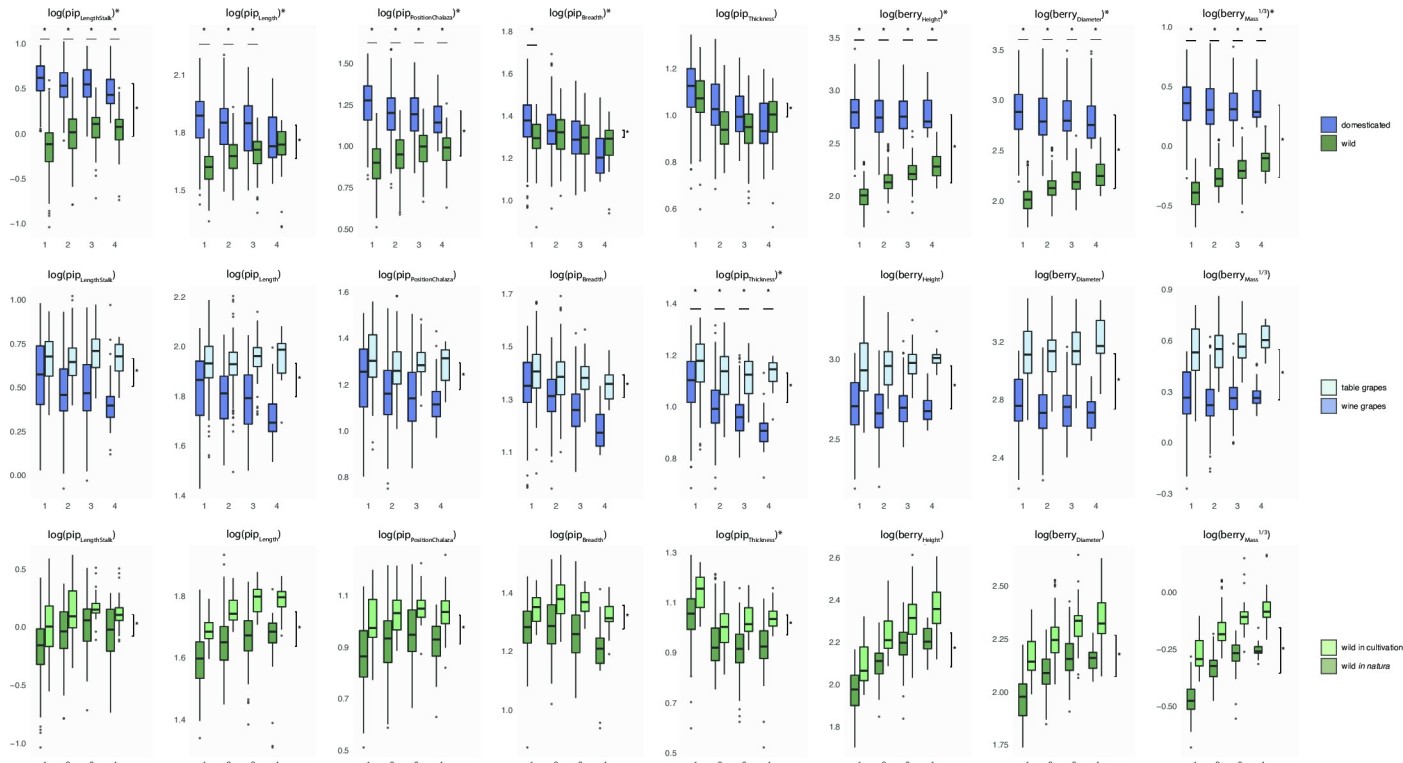

**Fig 2. Comparisons of (logged) lengths and (log cubic-rooted) mass measurements.** Each row represents a different comparison: a) wild and domesticated grapevines, b) table and wine varieties for domesticated accessions only, and c) cultivated and collected from wild for wild grapevine only. For each measurement, boxplots are displayed for each piposity level. Differences are tested using multivariate analyses of covariances, and differences of $P<10^{-5}$, are indicated by stars in the facet title (interaction), on the right (overall difference) and above each piposity level (difference within a given piposity).

$P<10^{-9}$; 3-pips: P = 0.016; 4-pips: P = 0.035 –Fig 4B). No differences within wild/domesticated and between successive piposities were found significant.

Mean shapes (Fig 5) illustrate these results. The mean absolute difference (MD) confirms that larger changes between extreme piposities are observed within wild grapevines (particularly for cultivated ones) and reveals that most of these changes affect the dorsal side of the pips (Fig 5).

## Pip shape and size in relation to status, accession and piposity; consequences for archaeobotanical inference

The respective contributions of berry height, accession and piposity on the shape of pips (Fig 6) show that the accession is the factor affecting the most the pip shape. The accession factor has a higher impact on domesticated grapevine than on wild, and on cultivated wild accessions than on those collected *in natura*. By contrast, its contributions for wine and table domesticated varieties were similar. Here again, piposity and berry height both affect the pip shape of wild accessions, but have a limited (piposity) and very limited (berry height) contribution for domesticated accessions.

Classification accuracies were compared using different training data and on different subsets (Fig 7). When different piposity levels were pooled, mirroring archaeobotanical admixtures, classification was very good at the status level (Fig 7A). Size + shape performed better (95%), than shape (93.7%) and size (92.5%) alone. When these models were evaluated on

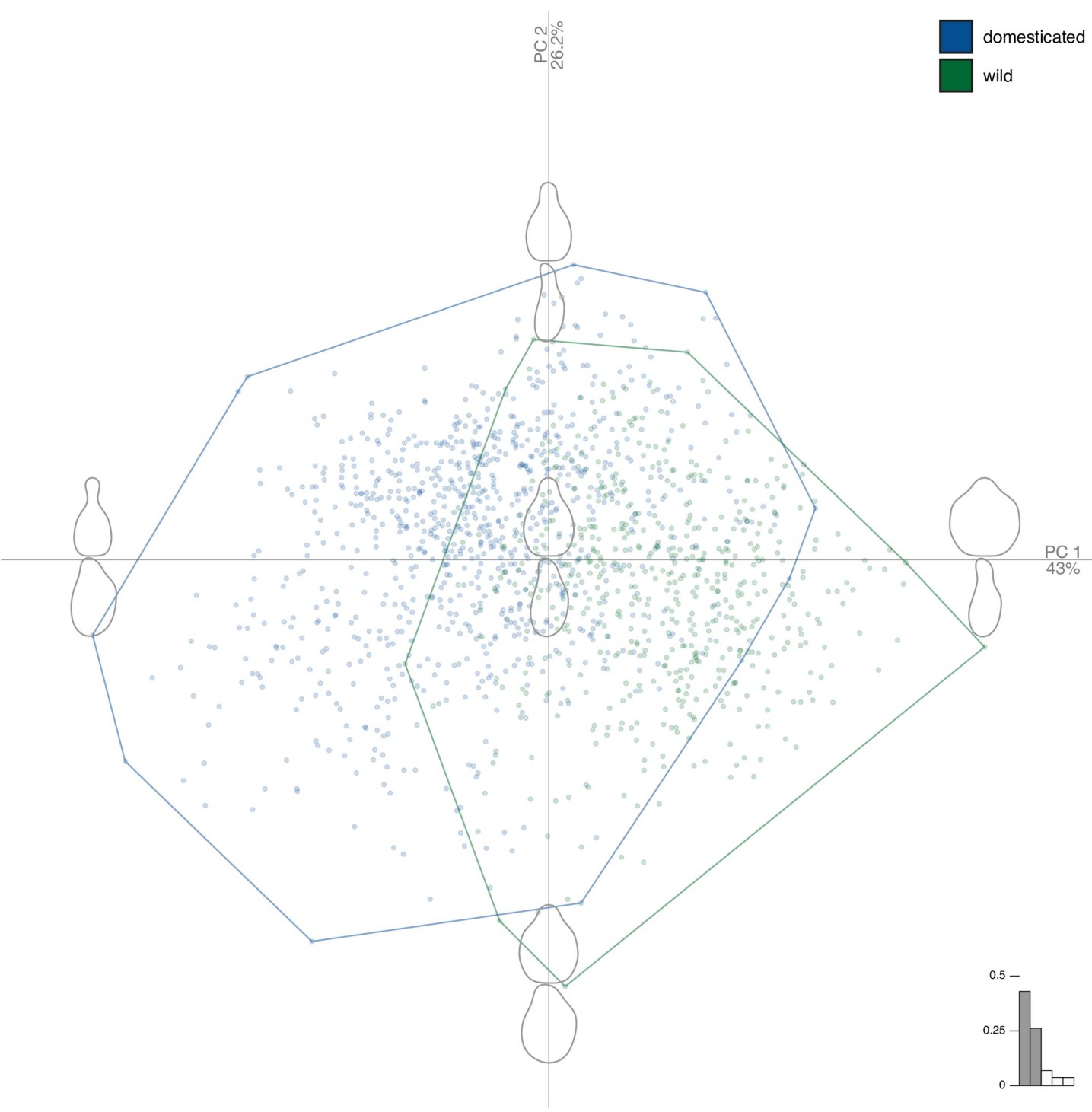

**Fig 3. Principal component analysis on the joint matrices of Fourier coefficients obtained for the two views.** The first two principal component gathered 69% of the total variance. The component of shape variation they capture are illustrated with reconstructed shapes at each extreme of their range. Colour of markers and convex hulls indicate pips from wild (green) and domesticated (blue) grapevines.

piposity subsets, they all have an accuracy above 91%, except for 4-pips berries. As expected, accuracies were lower at the accession level (Fig 7B) and when piposity levels were pooled, size

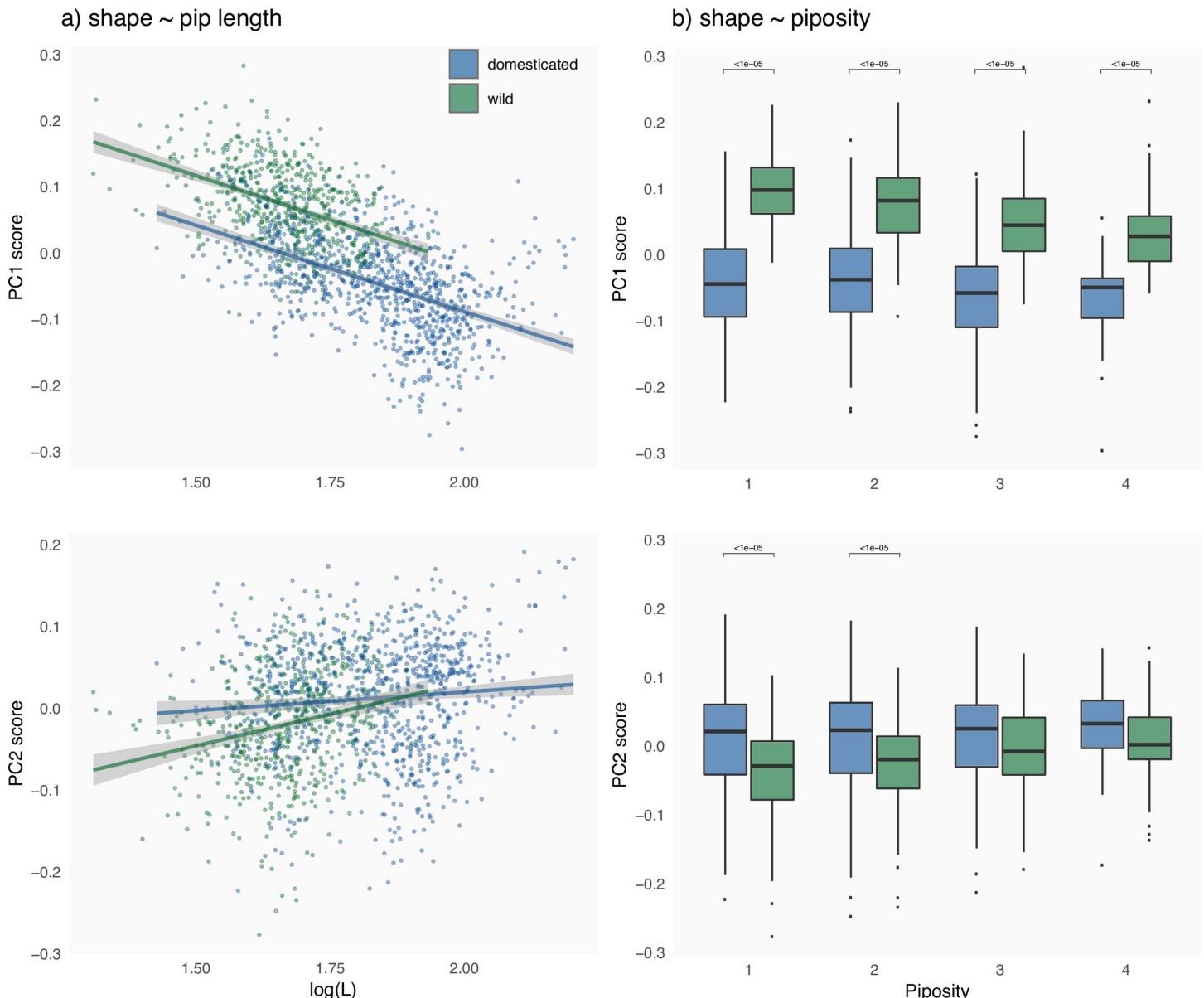

**Fig 4.** a) Regressions PC1 and PC2 versus pip length for domesticated pips (blue) and cultivation (green); b) boxplots for each piposity level for domesticated pips (blue) and cultivation (green). Differences are tested using Wilcoxon rank tests, and differences of $P<10^{-5}$, are indicated by brackets above graphs.

+ shape (89.8%) outperformed shape alone (81.3%) and size alone (46.3%). The same model ranking was observed on piposity subset, except for 4-pips berries. Overall, accuracies obtained were much higher than chance alone.

## Application to archaeological pips: Can we infer the dimensions of the (vanished) berry dimensions from the (recovered) pips?

On modern material, we used the size of pips to predict berry heights and diameters. Both regressions show a significant interaction of the domestication status (berry$_{Diameter}$: df = 1, F = 8369, P$<10^{-16}$; berry$_{Diameter}$: df = 1, F = 7730, P$<10^{-16}$), and two regressions for berry

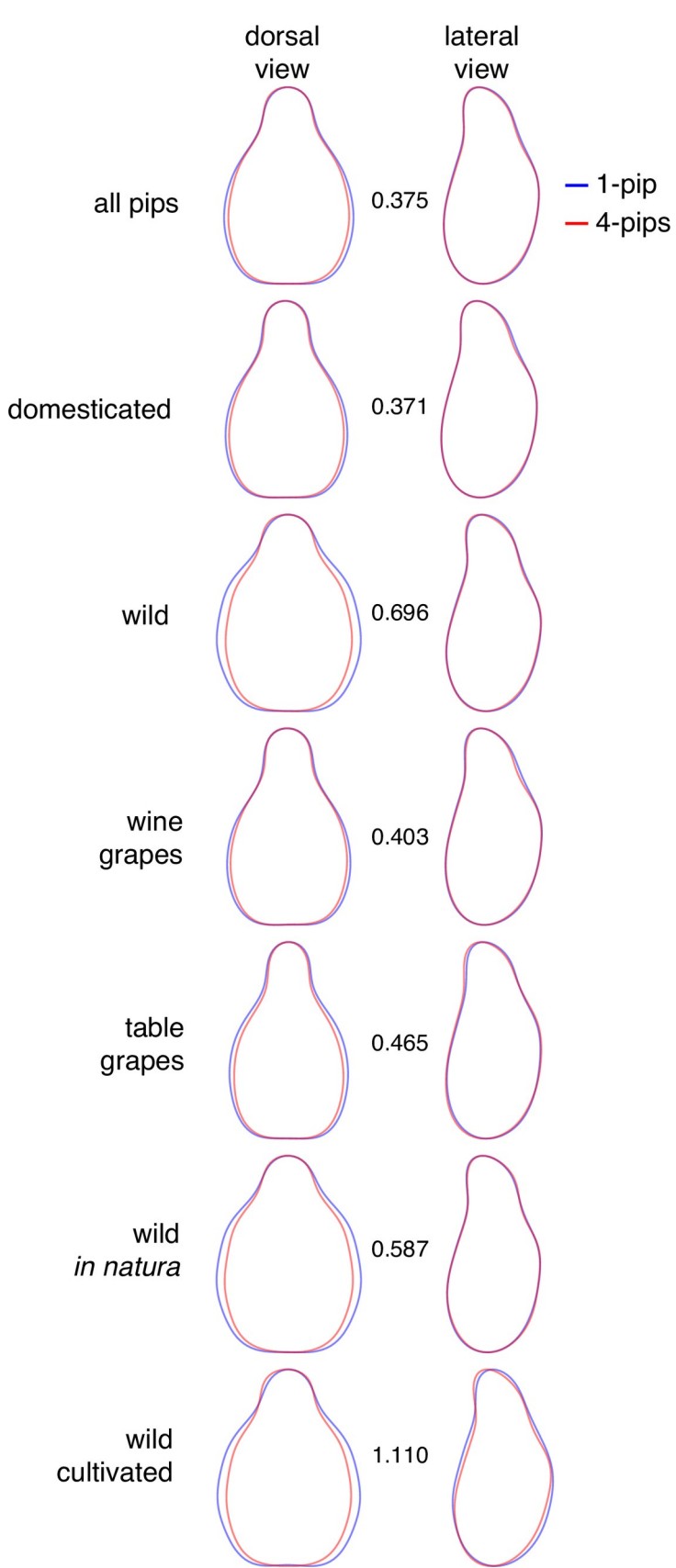

**Fig 5.** Mean shapes calculated for all pips sampled from berry with 1 (blue) or 4 pips (red), for different subsets (in rows) and for the dorsal and lateral views (in columns). Between the two views, an index of shape differences between these two extreme piposity levels (0 = identical shapes; unit = as much differences as between wild and domesticated average shapes).

diameter and two others for its height were obtained (Fig 8). All were significant (all $P<10^{-16}$) yet the adjusted $r^2$ were low (berry$_{Diameter}$ adj. $r^2_{wild}$ = 0.585, adj. $r^2_{domesticated}$ = 0.491; berry$_{Height}$ adj. $r^2_{wild}$ = 0.615, $r^2_{domesticated}$ = 0.511). Final models all used pip$_{Length}$, pip$_{Thickness}$, and at least one PC. Table 1). On unlogged berry diameter and height, the relative deviations were obtained (S3 Fig). Mean relative deviation per accession for berry$_{Diameter}$ ranged from -12.9% to +10.3% for wild, and from -22.9% to +17.7% for domesticated; for berry$_{Height}$ they ranged from -13.0% to +13.1% for wild, and from -29.4% to +29.4% for domesticated. The average predictions were all centred (on zero) ±1.6%.

Then, these four models were applied on the archaeobotanical material after being classified at the wild/domesticated level using LDA. 46 pips (22%) were classified with a posterior probability <0.8 and were filtered out. Among the remaining pips, 114 (72%) were classified as domesticated and 45 (28%) as wild. When compared to their modern analogues (Fig 9), the length of "domesticated pips" were closer to those of wine varieties than table varieties; the lengths of "wild pips" were intermediate between wild accessions collected in their habitat and those cultivated. For archaeological pips identified as domesticated, both inferred berry height and diameter were intermediate between wine and table modern varieties yet closer to wine ones. Similarly, for wild archaeological material, inferred berry height and diameter were intermediate between wild collected in their habitat and those grown in collection.

## Discussion

This study opens new fronts in our understanding of *Vitis vinifera* phenotypic changes under domestication and helps disentangle the interplay of the number of pips per berry, berry dimensions, domestication, pip shape, varietal diversity and cultivation practices in both wild and domesticated grapevines. We discuss implications for *Vitis vinifera* eco-evo-devo and perspectives for archaeobotanical studies for which a possible application is proposed.

### Patterns of covariation between the form of the pip, the form of the berry and the piposity

For grapevine, the theoretical maximum number of pips per berry is four, yet one was observed with five. Such abnormal piposity has already been reported [8, 36]. More than 70% berries had only one or two pips which is in accordance with previous publication [35]. There were no differences in piposity neither between domesticated or wild (Fig 1), nor between cultivated wild and those collected *in natura*.

Wild pips and berries are smaller than their domesticated counterparts; those from wine varieties are smaller than those of table varieties; and those of wild grapevines collected *in natura* are also smaller than those from cultivated wild individuals. This study details the effect of piposity on the pip form reported by previous studies [10, 24, 34, 35]. Among vertebrate dispersed plants, the reward (the fruit pulp mass) associated with a given seed mass is commensurate with work required to move it, and is expected to scale relatively [37]. For wild grapevine the berry and pip dimensions are thus expected to be constrained by their dispersers and by a general trade-off between pip size and number [38].

For all but wine varieties, the higher the piposity the longer the pip and the bigger the berry in which they develop (Fig 2). For these groups, it seems that more numerous pips are not

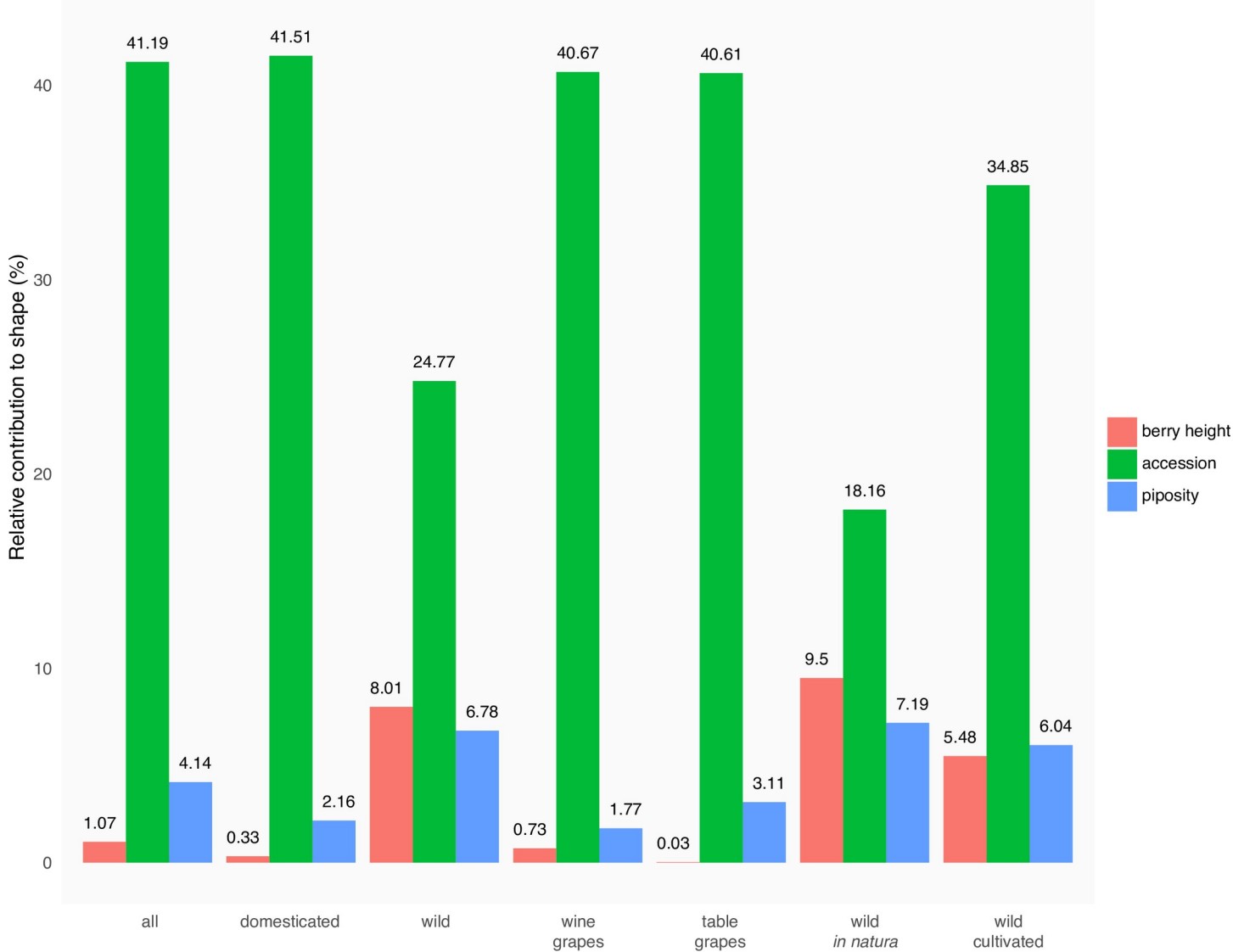

**Fig 6. Relative contribution of berry height, accession and number of pips per berry (coloured bars) onto the shape of pips for different subsets.**

limited by space or nutrients but rather contribute the development of bigger berries. The two stages of berry development are well known [39, 40]. The first, prior to anthesis, is a period of rapid berry growth mostly due to cell division. After anthesis, berry growth is largely due to cell enlargement and it has been suggested that pip growth may also increases cell mitosis in the developing berry [41]. Auxins, cytokinins and gibberelins, upregulated shortly after fertilisation in grapevine ovaries, are likely to trigger berry growth by cell expansion [35].

The absence of positive (or even negative) correlations between piposity, pip and berry dimensions for wine varieties remains unclear. For these varieties, the regulation, if any, may be at the bunch or stock scale, whether it has been selected (for example to concentrate sugars, aromas and flavours) or it is a by-product of another trait under selection. Since table varieties are larger than wine varieties, the berry dimensions of the latter cannot be argued to have reached a developmental limit.

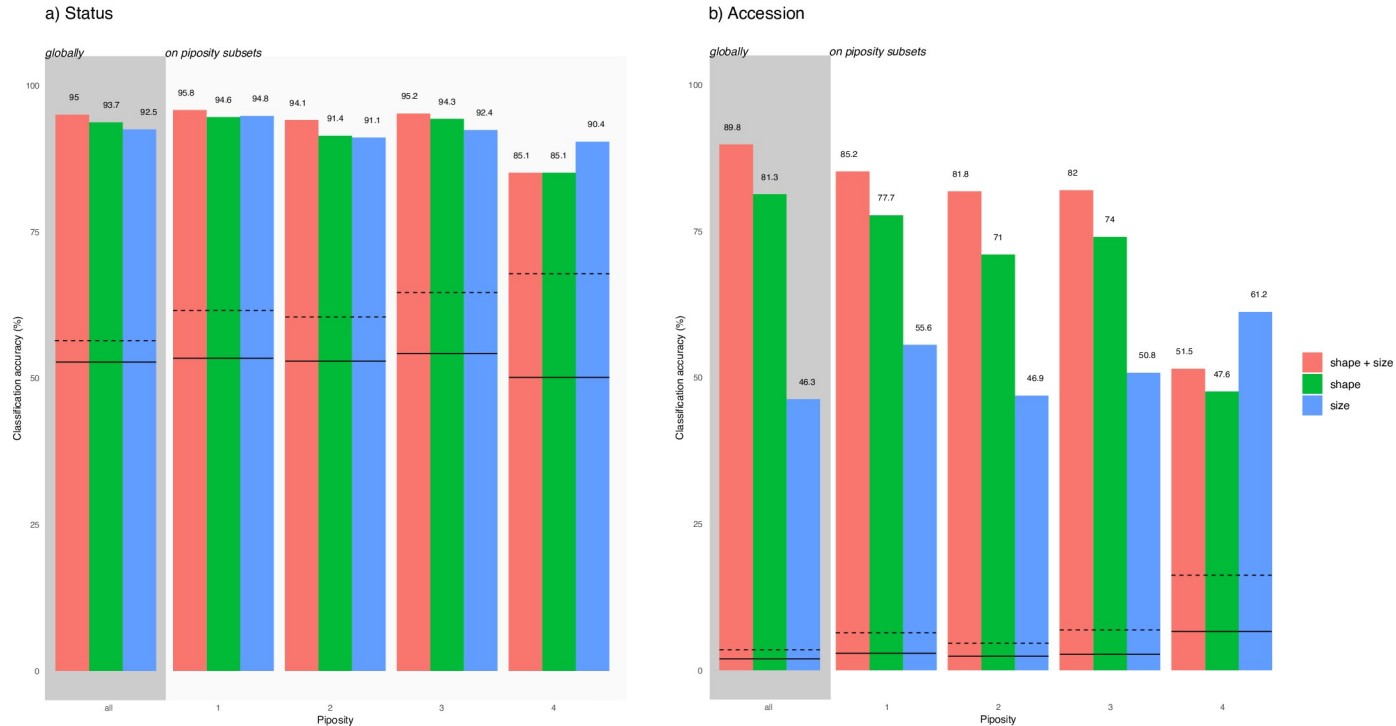

**Fig 7. Classification accuracy (LDA leave-one-out) at the a) status and b) accessions levels.** Models are trained (and evaluated) on the admixture of pips, then evaluated on piposity subsets. Different combinations of training data are used (Fourier coefficients of shape, lengths/mass measurements, both). Lines provide a random baseline and summarise 10000 permutations: solid line correspond to mean accuracy; dashed line to the maximal values obtained.

Finally, bivariate correlations concerning berry dimensions and mass are the strongest observed. This indicate robust allometries between berry size and mass, in other words that berry largely remain ellipsoid in shape, independently of their dimensions.

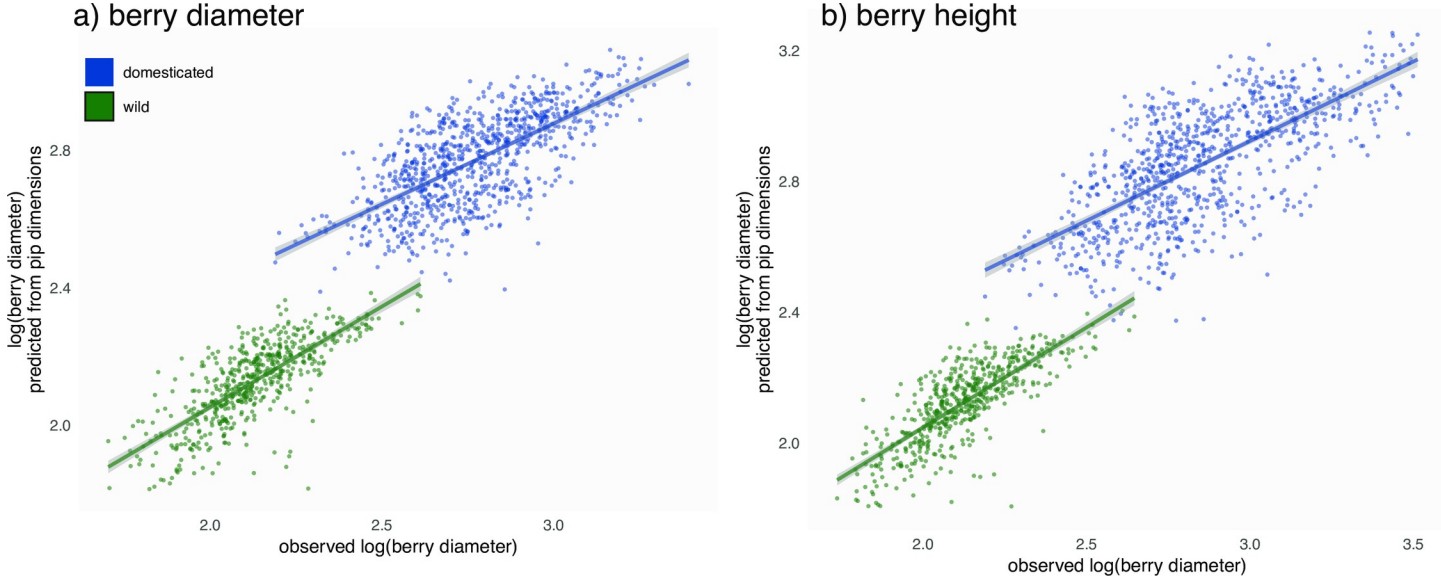

**Fig 8. Regressions for berry dimensions from pip dimensions, obtained on modern material: Predicted versus actual (logged) berry dimensions at the domestication status level (all accessions).** Columns are for berry diameter and height, respectively.

**Table 1. Estimates for pip lengths used to infer berry dimensions.**

| log Predicted: | subset | Intercept | log pip$_{Length}$ | log pip$_{LengthStalk}$ | log pip$_{PositionChalaza}$ | log pip$_{Breadth}$ | log pip$_{Thickness}$ | PC1 | PC2 |
|---|---|---|---|---|---|---|---|---|---|
| berry$_{Diameter}$ | wild | 0.689 | 1.136 | – | – | – | -0.438 | -0.223 | 0.209 |
| | domesticated | 1.298 | 0.478 | – | 0.149 | -0.150 | 0.520 | – | 0.430 |
| berry$_{Height}$ | wild | 0.838 | 0.732 | – | 0.216 | 0.171 | -0.318 | -0.524 | 0.258 |
| | domesticated | 0.760 | 1.312 | 0.279 | – | -0.783 | 0.584 | 1.021 | – |

Variables were all logged; so that berry$_{Diameter}$ (in mm and for wild) can be obtained with exp[0.65163 +1.19914×log(pip$_{Length}$) +0.10617×log(pip$_{PositionChalaza}$) -0.13263×log(pip$_{Breadth}$) -0.45449×log(pip$_{Thickness}$)].

## Morphometrics and domestication as a wedge into grapevine eco-evo-devo

For grapevine and domesticated plants in general, domestication results in a change of desirable phenotypic patterns but also releases many "natural" constraints such as

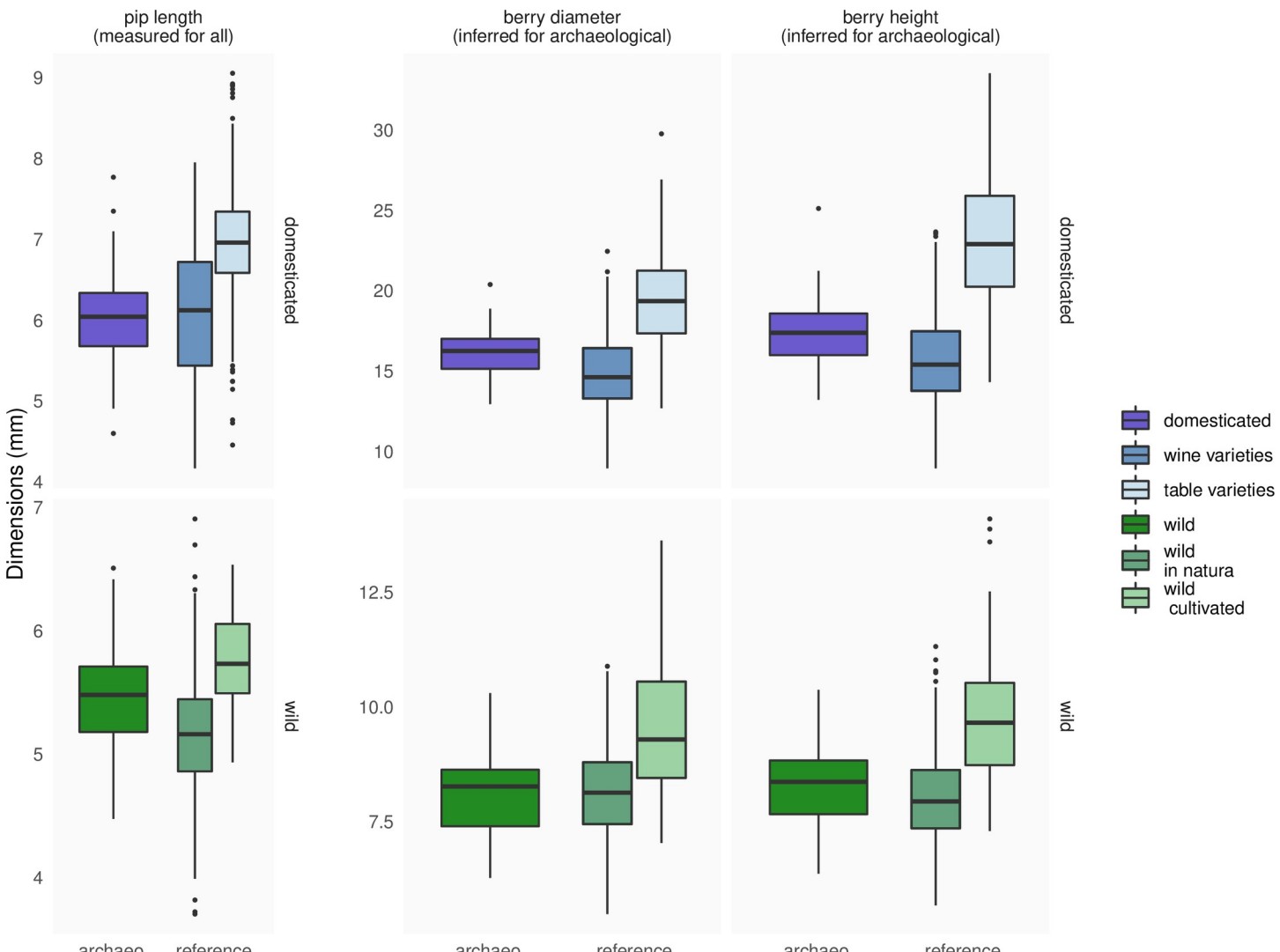

**Fig 9. Distribution of pip lengths (observed for all), berry height and diameter inferred from pip dimensions for the archaeological material from Sauvian–La Lesse.** Rows distinguish domesticated and wild grapes since separate regressions were required.

dispersion [42]. Cultivation practices such as pruning may explain why wild individuals grown in collection have bigger berries for higher piposity: the number of bunches is reduced, leading to larger pips. Cultivation also reduces growth constraints such as competition for water and light, self-supporting and climbing costs, those related to dispersers, etc.

Evidence of plastic and canalized phenotypic expression may be fuel for further eco-evo-devo studies. The latter brings a conceptual and experimental framework that relies on environmentally mediated regulatory systems to better understand ecological and evolutionary changes [43]. Here, the norm of reaction of the pip size and shape, along increasing piposity and berry dimensions, is clearly different at the three investigated levels: between wild and domesticated, between wine and table, between wild individuals grown in collection and those collected in their natural habitat.

### Consequences for archaeological inference

Some pip lengths differences between wild and domesticated appear more "robust" to increasing piposity, notably $pip_{LengthStalk}$ and $pip_{PositionChalaza}$ (Fig 2A; and S1 Fig). The use of lengths in discriminating between wild and domesticated types has long been used, including when archaeological material is charred [21, 32, 33]. Shape is confirmed here to refine such identification [22, 24]. We show here that shape is also affected by piposity for wild accessions (Fig 4). Most of these changes affect the dorsal side of wild pips, particularly when piposity is high. There is as much as ~70% difference, between 1- and 4-pips for wild accessions, than those observed between wild and domesticated types. Differences in extreme piposity are even larger than this "domestication gap", for the cultivated wild. This does not answer the question whether past vineyards cultivated "true" wild grapevines or "weakly" domesticated forms [24, 25] but it nevertheless points out how piposity and cultivation practices may contribute to this confusion by enhancing the continuum of pip forms.

Pip shape being largely used in archaeobotany, it was crucial to point out which factors contribute to its variability and if they could preclude archaeobotanical. We show that the main factor associated to pip differences was, by far, the accession and it was even more important for domesticated accessions. In other words, accession effect appears stronger than domestication (Fig 6). Comparatively, berry dimensions and piposity poorly contributed to shape differences. This confirms the usefulness and robustness of shape to identify shape archetypes. It may also indicate that domestication favours pip shape diversification whether this results from genetical linkage or drift.

Classification accuracies at the status level were all good (Fig 7). Shape was superior to size but, when considered jointly, the accuracies was improved. Whenever possible, both should thus be included. Concerning the accession level, results evidenced even more clearly these conclusions. Finally, we show that piposity is very unlikely to affect archaeobotanical identification either at the status or accession levels.

### An application on archaeological material: Inferring berry size from pip

Berry is very likely home to the most selected traits, from the beginnings of domestication to varietal breeding and diversification times. Unfortunately, such fleshy parts are usually too degraded, or even absent, to be quantified in archaeological material. The only route is thus inference based on modern material. Here, multiple regressions on pip dimensions show that berry diameter (Fig 8A) and height (Fig 8B) were centred and in the ±25% range.

Our archaeological application used material from Sauvian—La Lesse, a Roman farming establishment involved in wine production, were an admixture of wild and domesticated type

is attested [44]. Berry dimensions inferred from pips are intermediate between the wild growing *in natura* and those cultivated (Fig 9). This may suggest that wild, or weakly domesticated, individuals were cultivated in Roman vineyards. The berry dimensions inferred for domesticated varieties were closer to modern wine varieties than to table ones. This is congruent with the wine production attested at this period and in this region [44, 45].

## Conclusion

The main finding of this exploration of berry and pip form covariation is that for wild grapevine, the higher the piposity, the bigger is the berry and the longer is the pip. For both wild and domesticated, the longer is the pip, the more its shape looks like "domesticated". Further studies and dedicated experimental plans will help clarify the contribution of cultivation practices and, more generally on the interplay of genetic, functional and evolutionary changes that occurred in *Vitis vinifera* between the pip, its reproductive unit, and the berry, its dispersal reward and the main target of its domestication and varietal improvement.

## Material and methods

### Statistical environment

Statistical analyses were performed using the R 3.6.2 environment [46], the package Momocs 1.3.0 for everything morphometrics [47] and the tidyverse 1.2.1 packages for data manipulation and most graphics [48]. Alpha significance level was chosen equal to $10^{-5}$ all along analyses. This level both ensure marked differences for subsequent archaeobotanical application and an overall alpha level below 0.05 when repeated tests were done.

### Nomenclature

*Status* designates compartment (domesticated vs. wild); *accession* designates the variety/individual for domesticated/wild grapevine; synecdochically, a domesticated/wild pip/berry refers to the accession they were collected from; *cultivation* designs whether wild individuals were grown in field collection or sampled *in natura*; *form* is used when *shape* and *size* are used in combination; "piposity" is short for "given a pip, the number of pips in the berry where it was sampled".

### Modern and archaeological material

The modern reference material included 49 accessions (30 domesticated and 19 wild) from Euro-Mediterranean traditional cultivars and wild grapevines (Table 2). Fourteen wild grapevines were collected at ripeness in their habitat, and five were cultivated in the French central ampelographic collection (INRA, Vassal-Montpellier Grapevine Biological Resources Center; https://www6.montpellier.inra.fr/vassal), along with the domesticated accessions, which filters out most of environmental confounding effects such as different soils, climate or cultivation practices. Of the domesticated accessions, 21 were wine varieties and 9 table varieties. For each accession, 30 normally developed berries have been haphazardly collected from a single, fully ripe bunch.

Archaeobotanical material comes from two wells at the Roman farm of Sauvian—La Lesse, extensively described elsewhere (US3022, US3063, US3171 and US3183 in [44]). These archaeological layers were dated to 2025–1725 BP based on pottery and coins. The waterlogged conditions ensured very good preservation of the pips (N = 205).

**Table 2. Accessions used in this study.**

| id | accession | status | dest./cult. | $pip_{Length}$ | $pip_{LengthStalk}$ | $pip_{PositionChalaza}$ | $pip_{Breadth}$ | $pip_{Thickness}$ | $berry_{Height}$ | $berry_{Diameter}$ | $berry_{Mass}$ |
|---|---|---|---|---|---|---|---|---|---|---|---|
| cAlvarB | Alvarinho | dom. | wine | 5.18 ± 0.28 | 1.35 ± 0.19 | 2.66 ± 0.23 | 3.5 ± 0.21 | 2.5 ± 0.15 | 10.6 ± 0.81 | 10.93 ± 0.98 | 0.82 ± 0.21 |
| cBarbeN | Barbera | dom. | wine | 6.73 ± 0.33 | 1.95 ± 0.14 | 3.58 ± 0.2 | 3.67 ± 0.23 | 2.7 ± 0.14 | 16.38 ± 1.07 | 15 ± 1.07 | 2.44 ± 0.49 |
| cCabSaN | Cabernet-Sauvignon | dom. | wine | 5.93 ± 0.34 | 1.78 ± 0.16 | 3.09 ± 0.2 | 3.57 ± 0.22 | 2.65 ± 0.17 | 14.05 ± 0.89 | 13.47 ± 0.84 | 1.57 ± 0.27 |
| cCarigN | Carignan | dom. | wine | 6.85 ± 0.26 | 2.14 ± 0.21 | 3.85 ± 0.24 | 3.54 ± 0.22 | 2.73 ± 0.09 | 18.1 ± 0.83 | 16.85 ± 0.84 | 3.16 ± 0.41 |
| cChardB | Chardonnay | dom. | wine | 5.28 ± 0.41 | 1.38 ± 0.2 | 3.06 ± 0.28 | 3.72 ± 0.28 | 2.78 ± 0.27 | 13.42 ± 1.24 | 12.9 ± 1.16 | 1.69 ± 0.41 |
| cChevkN | Chevka | dom. | wine | 5.97 ± 0.33 | 1.48 ± 0.13 | 3.05 ± 0.19 | 3.73 ± 0.19 | 2.53 ± 0.1 | 18.08 ± 1.17 | 16.51 ± 1.11 | 3.09 ± 0.58 |
| cDebinB | Debina | dom. | wine | 7.01 ± 0.48 | 1.73 ± 0.21 | 3.7 ± 0.3 | 4.21 ± 0.28 | 3.41 ± 0.19 | 19.61 ± 1.9 | 17.67 ± 1.58 | 3.89 ± 0.87 |
| cFesAlB | Feteasca alba | dom. | wine | 5.19 ± 0.23 | 1.4 ± 0.13 | 2.65 ± 0.19 | 3.46 ± 0.19 | 2.5 ± 0.13 | 12.49 ± 0.79 | 12.99 ± 0.69 | 1.62 ± 0.26 |
| cGaidoB | Gaïdouria | dom. | wine | 5.55 ± 0.37 | 1.61 ± 0.19 | 2.92 ± 0.23 | 3.8 ± 0.23 | 2.94 ± 0.14 | 16.51 ± 1.36 | 17.1 ± 1.3 | 3.46 ± 0.73 |
| cGrenaN | Grenache | dom. | wine | 5.32 ± 0.29 | 1.46 ± 0.1 | 2.84 ± 0.17 | 3.38 ± 0.2 | 2.45 ± 0.14 | 15.91 ± 0.78 | 15.12 ± 0.66 | 2.3 ± 0.27 |
| cKypreN | Kypreiko | dom. | wine | 6.79 ± 0.39 | 2.05 ± 0.23 | 3.81 ± 0.26 | 3.99 ± 0.27 | 3.07 ± 0.12 | 21.38 ± 1.35 | 17.98 ± 1.16 | 4.26 ± 0.79 |
| cMavruN | Mavrud | dom. | wine | 6.9 ± 0.26 | 2.35 ± 0.18 | 3.93 ± 0.2 | 4.3 ± 0.19 | 3.19 ± 0.14 | 14.91 ± 0.9 | 14.53 ± 0.84 | 2.04 ± 0.31 |
| cMerloN | Merlot | dom. | wine | 6.02 ± 0.32 | 1.5 ± 0.15 | 3.01 ± 0.24 | 3.77 ± 0.24 | 2.75 ± 0.18 | 12.62 ± 1.04 | 12.83 ± 1.16 | 1.41 ± 0.31 |
| cMesFrB | Meslier Saint François | dom. | wine | 5.51 ± 0.59 | 1.53 ± 0.2 | 3.05 ± 0.34 | 3.27 ± 0.31 | 2.61 ± 0.29 | 16.59 ± 1.64 | 15.54 ± 1.55 | 2.86 ± 0.77 |
| cMourvN | Mourvèdre | dom. | wine | 6.17 ± 0.41 | 1.51 ± 0.18 | 3.11 ± 0.27 | 3.73 ± 0.28 | 2.89 ± 0.2 | 15.71 ± 1.04 | 15.27 ± 1.16 | 2.28 ± 0.49 |
| cMusPGRs | Muscat à petits grains roses | dom. | wine | 5.23 ± 0.3 | 1.33 ± 0.19 | 2.95 ± 0.23 | 3.12 ± 0.18 | 2.39 ± 0.13 | 12.93 ± 1.16 | 13.82 ± 0.9 | 1.97 ± 0.28 |
| cMusPGN | Muscat noir à petits grains | dom. | wine | 7.31 ± 0.29 | 2.27 ± 0.19 | 4.18 ± 0.26 | 4.2 ± 0.25 | 3.3 ± 0.16 | 20.07 ± 1.3 | 18.99 ± 1.39 | |
| cPinNoN | Pinot noir | dom. | wine | 6.35 ± 0.31 | 1.77 ± 0.21 | 3.54 ± 0.2 | 3.93 ± 0.4 | 2.78 ± 0.19 | 13.87 ± 1.05 | 13.42 ± 1.19 | 1.83 ± 0.39 |
| cRoussB | Roussanne | dom. | wine | 6.57 ± 0.23 | 1.46 ± 0.15 | 3.39 ± 0.19 | 4.16 ± 0.23 | 3.09 ± 0.2 | 15.37 ± 0.77 | 14.91 ± 0.83 | 2.13 ± 0.32 |
| cSauviB | Sauvignon | dom. | wine | 6.53 ± 0.39 | 1.91 ± 0.19 | 3.49 ± 0.28 | 3.74 ± 0.18 | 2.68 ± 0.16 | 15.46 ± 1.07 | 13.45 ± 0.87 | 1.78 ± 0.3 |
| cSyrahN | Syrah | dom. | wine | 5.39 ± 0.35 | 1.64 ± 0.18 | 3.14 ± 0.28 | 3.21 ± 0.17 | 2.54 ± 0.15 | 15.37 ± 0.91 | 13.51 ± 0.85 | 1.99 ± 0.29 |
| cAinBoB | Ain el Bouma | dom. | table | 5.56 ± 0.41 | 1.9 ± 0.2 | 3.42 ± 0.23 | 3.53 ± 0.29 | 2.63 ± 0.14 | 21.11 ± 2.06 | 17.96 ± 1.8 | 4.51 ± 1.1 |
| cChaBlB | Chaouch blanc | dom. | table | 6.95 ± 0.26 | 1.74 ± 0.16 | 3.86 ± 0.31 | 4.08 ± 0.25 | 3.44 ± 0.21 | 24.67 ± 1.85 | 22.55 ± 1.82 | 7.76 ± 1.59 |
| cFerTaR | Ferral tamara | dom. | table | 6.79 ± 0.25 | 1.95 ± 0.18 | 3.52 ± 0.17 | 4.21 ± 0.18 | 3.29 ± 0.16 | 23.45 ± 1.22 | 19.73 ± 1.33 | 5.79 ± 0.83 |
| cHadarB | Hadari | dom. | table | 7.42 ± 0.38 | 1.89 ± 0.19 | 3.49 ± 0.28 | 4.2 ± 0.19 | 3.24 ± 0.15 | 21.54 ± 1.2 | 19.37 ± 1.03 | 4.73 ± 0.7 |
| cHunisN | Hunisa | dom. | table | 8.32 ± 0.39 | 2.27 ± 0.23 | 4.41 ± 0.27 | 4.94 ± 0.25 | 3.4 ± 0.2 | 30.28 ± 2.53 | 23.92 ± 1.68 | 9.87 ± 1.48 |
| cKarPaRs | Kara papigi | dom. | table | 6.58 ± 0.36 | 1.92 ± 0.2 | 3.21 ± 0.29 | 3.65 ± 0.22 | 2.8 ± 0.18 | 18.45 ± 1.67 | 16.29 ± 1.33 | 3 ± 0.73 |
| cKraTzN | Kravi tzitzi | dom. | table | 7 ± 0.31 | 1.92 ± 0.21 | 3.51 ± 0.22 | 3.96 ± 0.17 | 3.09 ± 0.14 | 28.02 ± 2.11 | 19.37 ± 1.43 | 6.43 ± 1.19 |
| cPemGeR | Pembe Gemre | dom. | table | 6.98 ± 0.33 | 2.11 ± 0.18 | 3.93 ± 0.26 | 3.84 ± 0.18 | 2.91 ± 0.09 | 16.46 ± 0.83 | 16.16 ± 1.02 | 3.07 ± 0.46 |
| cSlivaN | Sliva | dom. | table | 6.98 ± 0.47 | 1.95 ± 0.29 | 3.67 ± 0.35 | 4.02 ± 0.2 | 3.23 ± 0.11 | 23.41 ± 1.65 | 19.8 ± 1.74 | 5.59 ± 1.28 |
| wCamSa4 | Camp Saure | wild | in natura | 5.23 ± 0.26 | 1.04 ± 0.15 | 2.46 ± 0.17 | 3.38 ± 0.17 | 2.5 ± 0.2 | 8.29 ± 0.83 | 8.15 ± 1.02 | |
| wChala7 | Chalabre | wild | in natura | 4.83 ± 0.28 | 0.92 ± 0.14 | 2.29 ± 0.17 | 3.59 ± 0.16 | 2.52 ± 0.25 | 7.68 ± 0.75 | 8.21 ± 0.92 | |
| wCoBab2 | Col de la Babourade | wild | in natura | 4.8 ± 0.56 | 0.91 ± 0.22 | 2.41 ± 0.31 | 3.24 ± 0.32 | 2.31 ± 0.27 | 8.26 ± 0.96 | 8.56 ± 0.93 | |
| wEscal13 | L'Escale (13) | wild | in natura | 5.51 ± 0.38 | 1.07 ± 0.18 | 2.84 ± 0.22 | 3.58 ± 0.2 | 2.76 ± 0.2 | 9.44 ± 0.97 | 9.14 ± 1.06 | |
| wEscal14Bis | L'Escale (14) | wild | in natura | 4.84 ± 0.32 | 0.82 ± 0.12 | 2.24 ± 0.18 | 3.32 ± 0.15 | 2.4 ± 0.17 | 7.72 ± 0.81 | 7.81 ± 0.95 | 0.32 ± 0.1 |
| wEscal16 | L'Escale (16) | wild | in natura | 5.34 ± 0.28 | 1.18 ± 0.14 | 2.69 ± 0.17 | 3.6 ± 0.26 | 2.56 ± 0.23 | 7.74 ± 0.7 | 7.76 ± 0.92 | 0.35 ± 0.09 |
| wEscal17 | L'Escale (17) | wild | in natura | 4.94 ± 0.27 | 1.02 ± 0.16 | 2.38 ± 0.16 | 3.61 ± 0.18 | 2.77 ± 0.19 | 7.37 ± 0.68 | 7.41 ± 0.85 | 0.33 ± 0.09 |
| wEscal18 | L'Escale (18) | wild | in natura | 5.17 ± 0.46 | 0.72 ± 0.16 | 2.34 ± 0.24 | 3.95 ± 0.37 | 2.78 ± 0.33 | 7.45 ± 0.78 | 7.84 ± 0.8 | |
| wEscal20 | L'Escale (20) | wild | in natura | 5.05 ± 0.24 | 0.89 ± 0.14 | 2.39 ± 0.17 | 3.48 ± 0.2 | 2.53 ± 0.21 | 8.05 ± 0.67 | 8.23 ± 0.87 | 0.37 ± 0.1 |
| wCalme10 | La Calmette (10) | wild | in natura | 5.28 ± 0.31 | 0.89 ± 0.15 | 2.58 ± 0.23 | 3.7 ± 0.24 | 2.77 ± 0.27 | 7.98 ± 0.74 | 8.14 ± 0.82 | |
| wCalme11 | La Calmette (11) | wild | in natura | 4.93 ± 0.42 | 1.15 ± 0.25 | 2.66 ± 0.31 | 3.33 ± 0.24 | 2.59 ± 0.24 | 7.66 ± 1.09 | 7.57 ± 1.14 | |
| wPSL13 | Pic Saint Loup (13) | wild | in natura | 5.35 ± 0.29 | 0.98 ± 0.12 | 2.85 ± 0.22 | 3.75 ± 0.22 | 2.8 ± 0.3 | 8.75 ± 0.63 | 8.55 ± 0.73 | 0.42 ± 0.1 |
| wPSLH | Pic Saint Loup (H) | wild | in natura | 5.07 ± 0.44 | 0.75 ± 0.19 | 2.21 ± 0.27 | 3.78 ± 0.29 | 2.69 ± 0.26 | 7.67 ± 0.76 | 8.14 ± 0.88 | |
| wRivel1 | Rivel | wild | in natura | 5.83 ± 0.55 | 0.97 ± 0.27 | 2.77 ± 0.28 | 4.08 ± 0.28 | 3.24 ± 0.26 | 7.8 ± 0.74 | 7.88 ± 0.95 | 0.38 ± 0.11 |

(*Continued*)

**Table 2.** (Continued)

| id | accession | status | dest./cult. | pip$_{Length}$ | pip$_{LengthStalk}$ | pip$_{PositionChalaza}$ | pip$_{Breadth}$ | pip$_{Thickness}$ | berry$_{Height}$ | berry$_{Diameter}$ | berry$_{Mass}$ |
|---|---|---|---|---|---|---|---|---|---|---|---|
| cKetsc27 | Ile de Ketch (27) | wild | cultivated | 5.68 ± 0.27 | 1.08 ± 0.15 | 2.63 ± 0.2 | 3.86 ± 0.27 | 2.84 ± 0.19 | 8.88 ± 0.72 | 8.75 ± 0.71 | 0.59 ± 0.11 |
| cPalmA | Palma | wild | cultivated | 5.78 ± 0.32 | 1.51 ± 0.19 | 3.06 ± 0.2 | 3.72 ± 0.14 | 2.66 ± 0.23 | 11.27 ± 1.3 | 10.78 ± 1.44 | 0.91 ± 0.33 |
| cPSL13 | Pic Saint Loup (13) | wild | cultivated | 5.8 ± 0.34 | 0.91 ± 0.19 | 2.74 ± 0.25 | 4.12 ± 0.2 | 3.14 ± 0.25 | 8.79 ± 0.73 | 8.72 ± 0.96 | 0.52 ± 0.14 |
| cPSL5 | Pic Saint Loup (5) | wild | cultivated | 6.05 ± 0.17 | 1.14 ± 0.09 | 2.82 ± 0.13 | 3.9 ± 0.18 | 2.79 ± 0.13 | 10.67 ± 0.57 | 10.9 ± 0.65 | 0.77 ± 0.12 |
| wLambrN | wLambrN | wild | cultivated | 5.48 ± 0.3 | 1.06 ± 0.1 | 2.71 ± 0.2 | 3.85 ± 0.18 | 2.94 ± 0.25 | 9.12 ± 0.84 | 8.53 ± 0.96 | 0.54 ± 0.15 |

D: domesticated; W: wild; Wn: wine grape; Tb: table grape. For domesticated grapevines, names correspond to the variety names. Dimensions are reported with mean±sd and given in mm, except for berry$_{mass}$ which is expressed in g.

## Traditional measurements

On modern material, the berry diameter (berry$_{Diameter}$), height (berry$_{Height}$) and mass (berry$_{Mass}$) were obtained before dissection (Table 2). Mass was not available for 9 accessions that were removed from further analyses involving mass. Then, the number of pips (hereafter "piposity") was recorded and one pip was randomly chosen. A single berry from the variety "Kravi tzitzi" was found with 5 pips and was discarded from further analyses. The final dataset thus consisted of 1469 pips (48 accessions × 30 pips + 1×29).

All pips, were photographed in dorsal and lateral views by the same operator (TP) using an Olympus SZ-ET stereomicroscope and an Olympus DP camera. On each pip, five length measurements were recorded by the same operator (LB) using ImageJ [49] Table 2, Fig 10): total length (pip$_{Length}$), length of stalk (pip$_{LengthStalk}$), position of the chalaza (pip$_{PositionChalaza}$), breadth (pip$_{Breadth}$) and thickness (pip$_{Thickness}$). All length measurements were log-transformed to focus on relative changes and minimize size differences; the mass was log cubic-root transformed for the same reason [24].

As preliminary analyses on modern material, differences between average piposity were tested using generalized linear model with Poisson error; differences in their distributions were tested using two-sided Fisher's exact tests.

## Testing the covariation between pip and berry size in relation to the number of pips

On modern material, three sets of differences in pips and berries measurements were tested using multivariate analysis of covariance: i) the interaction between status and piposity; ii) if the latter was significant, we also tested differences between status for a given piposity level; iii) whether the average piposity differs between status. These three possible sets of differences were tested between different subsets: domesticated and wild accessions; wine and table varieties for domesticated accessions; cultivated wild individuals and those collected *in natura*. Piposity was then discarded and sets were compared using Wilcoxon rank tests.

Bivariate comparisons were explored between the domesticated and wild accessions (discarding piposity), and tested with an analysis of covariance. When the domestication status was significant, separate regressions were tested and, if significant, the adjusted r$^2$ was obtained.

## Testing the covariation between pip and berry shapes in relation to the piposity

For pips, shape data were extracted from the dorsal and lateral outlines. 2D coordinates were extracted from photographs, centred, scaled, aligned along their longer axis and normalized for the position of their first point. These preliminary steps removed positional, size, rotation and phasing differences between outlines before elliptical Fourier transforms (EFT). The latter were performed on the dorsal and lateral views separately, and the number of harmonics was chosen to gather 99% of the total harmonic power (8 for both views). This generated 64 coefficients for each pip (2 views × 8 harmonics per view × 4 coefficients per harmonic).

To explore the overall variability of shape, a principal component analysis (PCA) was calculated on the full matrix of coefficients. The first two PCs (see Results) were used as synthetic shape variables (Fig 3). To test the effect of piposity and pip dimension on pip shape, the same approach than for length measurements using PC1 and PC2 as the response variables. To test the relation between shape and pip length (only pip$_{Length}$ was used), analyses of covariance first tested if separate regressions were justified. Then Wilcoxon tests were used to test for shape differences between and within piposity levels.

## Dorsal view

## Lateral view

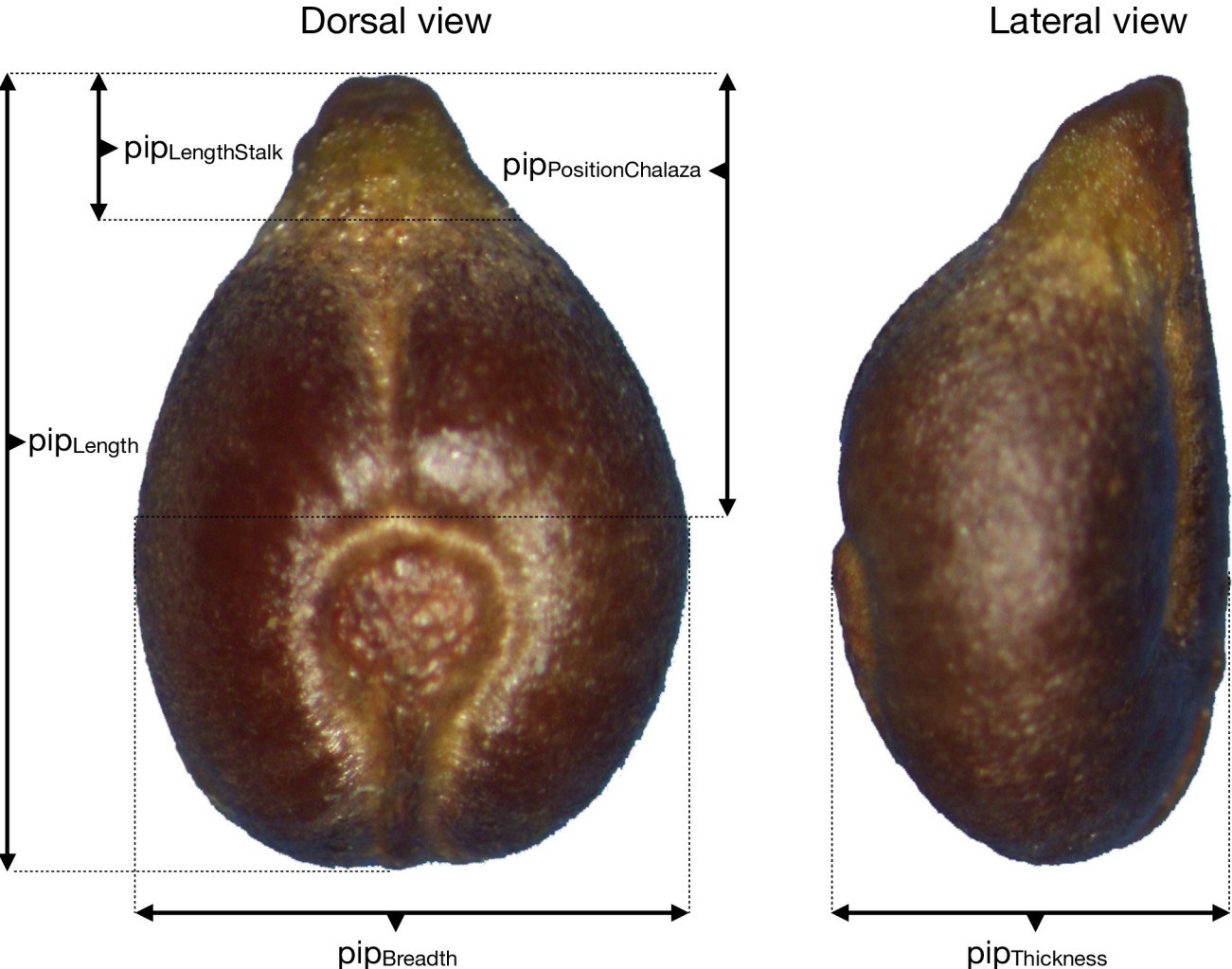

**Fig 10. Dorsal and lateral views of a grapevine pip, here from the *Vitis vinifera* subsp. *sylvestris* wild" individual "Pic Saint-Loup 13", with indications of morphometric measurements: pip$_{length}$ (total length), pip$_{stalk}$ (length of the stalk), pip$_{chalaza}$ (position of the chalaza), pip$_{width}$ and pip$_{thickness}$.**

To visualize shape differences between extreme piposity levels (1 and 4), mean shapes for the dorsal and lateral views were calculated on the matrix of coefficients. These differences were quantified with the mean absolute difference (MD) between each sets of Fourier coefficients. To make these differences meaningful, they were divided by the mean difference of Fourier coefficients between cultivated and wild accessions with all piposity levels pooled. For each subset, MD was calculated as:(| $\boxed{\text{coefficients}_{\text{subset, 1-pip}}}$ - $\boxed{\text{coefficients}_{\text{subset, 4-pips}}}$ |)/(| $\boxed{\text{coefficients}_{\text{domesticated, all pips}}}$ - $\boxed{\text{coefficients}_{\text{wild, all pips}}}$ |). For example, a MD equals to 0 indicates no difference between pips with a piposity of 1 or 4; a MD greater than unit indicates more differences relatively to differences that exist between domesticated and wild individuals.

### Pip shape and size in relation to status, accession and piposity

To quantify the respective contribution of berry dimensions, accession and piposity onto pip shape, a multivariate analysis of variance used the following model: all Fourier coefficients ~ berry$_{Height}$ + accessions + piposity within accession. Since it was highly correlated to other

berry measurements (see Results), only berry$_{Height}$ was used to describe berry dimensions. The contribution of each variable is the ratio of its sum of squares over the total sum of squares (including residuals). Again, this is tested on the different subsets of interest (Fig 6).

Linear discriminant analyses (LDA) were used to evaluate whether piposity could preclude status and accession classification accuracies. Different combinations of predictors (sizes; shape; sizes + shape) were evaluated to benchmark their performance to classify the pips to their correct status and accession (Fig 7).

Given a combination of (status, accession) × (sizes+shape, sizes, shape), a leave-one-out cross-validation was used to assess classification accuracies, evaluated on all pips, to mirror archaeological admixtures where piposity is unknown (Fig 7). To cope with unbalanced group structures, we calculated a baseline for each subset that estimates the mean and maximum accuracy one can obtain by chance, using $10^4$ permutations (see [50]). If the accuracy observed is higher than the maximum value obtained using permutations, the LDA can be considered to perform better than chance alone, with an estimated alpha below $10^{-4}$.

## Predicting the dimensions of the archaeological berry dimensions

Separate multivariate regressions were calculated on the modern material, for berry height and diameter, using the five length measurements on pips. As predictor variable, we used length measurements (for dimensions) and the first two principal components (for shape). The difference between domesticated and wild grapevines regressions was first tested using an analysis of covariance: two regressions (one for cultivated, one for wild) were obtained for the berry height and two others for its diameter (Fig 8). These four regressions were fitted using stepwise regression with backward elimination based on the AIC, and started with full models: berry$_{Height/Diameter}$ for wild/domesticated ~ pip$_{Length}$ + pip$_{LengthStalk}$ + pip$_{PositionChalaza}$ + pip$_{Breadth}$ + pip$_{Thickness}$ + PC1 + PC2, all but PCs were log-transformed). Then, archaeological pips were classified into domesticated or wild using an LDA trained using the same variables but of modern pips. Pips assigned to wild/domesticated with a posterior probability <0.8 were filtered out. Finally, the berry height and diameter of this archaeological material were inferred using the corresponding models (Fig 9).

## Supporting information

**S1 Fig. Comparisons of (logged) lengths and (cubic-rooted) mass measurements.** On rows are displayed different subsets: a) wild and domesticated grapevines, b) for domesticated accessions, table and wine varieties and, c) for wild accessions, those collected *in natura* and others cultivated as domesticated varieties. Different piposity levels are pooled (see Fig 3 for the detail). Differences are tested using Wilcoxon rank tests and all of them have a P<$10^{-5}$.
(TIF)

**S2 Fig. Bivariate pairwise plot between (logged) lengths and (cubic-rooted) mass measurements.** For the sake of readability, only the wild versus domesticated status are displayed using different colours (green for wild; blue for domesticated). If two regressions are justified, then they are shown using the corresponding colours; otherwise a single regression line is showed in black. Then, for each regression, the correlations are tested and, if significant, the adjusted $R^2$ is displayed on the regression lines.
(TIF)

**S3 Fig. Predictions obtained from regressions for berry dimensions from pip dimensions on modern material.** The relative deviation, at the accession level, and for unlogged measurements. Columns are for berry diameter and height, respectively.
(TIF)

## Acknowledgments

We thank the OSU-OREME (https://oreme.org/) that helped to the constitution of the wild grape pip collection. We are grateful to the INRA Vassal-Montpellier grapevine collection (Marseillan-Plage, France) that provided all the pips from cultivated varieties and cultivated wild grapes. We warmly acknowledge Michael Wallace for his help with English.

## Author Contributions

**Conceptualization:** Vincent Bonhomme, Sandrine Picq, Jean-Frédéric Terral, Laurent Bouby.

**Data curation:** Vincent Bonhomme, Sarah Ivorra, Thierry Pastor, Laurent Bouby.

**Formal analysis:** Vincent Bonhomme, Allowen Evin.

**Funding acquisition:** Laurent Bouby.

**Investigation:** Sandrine Picq, Isabel Figueiral, Jean-Frédéric Terral.

**Methodology:** Vincent Bonhomme, Sandrine Picq, Sarah Ivorra, Allowen Evin, Jean-Frédéric Terral, Laurent Bouby.

**Project administration:** Laurent Bouby.

**Resources:** Isabel Figueiral.

**Software:** Vincent Bonhomme.

**Supervision:** Roberto Bacilieri, Jean-Frédéric Terral, Laurent Bouby.

**Validation:** Thierry Lacombe, Jean-Frédéric Terral.

**Visualization:** Vincent Bonhomme.

**Writing – original draft:** Vincent Bonhomme.

**Writing – review & editing:** Vincent Bonhomme, Sandrine Picq, Sarah Ivorra, Allowen Evin, Roberto Bacilieri, Thierry Lacombe, Jean-Frédéric Terral, Laurent Bouby.

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
