## [Decision Letter · Decision Letter 0]

21 Jul 2020

PONE-D-20-10713

Eco-evo-devo implications and archaeobiological perspectives of trait covariance in fruits of wild and domesticated grapevines

PLOS ONE

Dear Sir

Thank you for submitting your manuscript to PLOS ONE. After careful consideration, we feel that it has merit but does not fully meet PLOS ONE’s publication criteria as it currently stands. Therefore, we invite you to submit a revised version of the manuscript that addresses the points raised during the review process.

Please submit your revised manuscript within 60 days. If you will need more time than this to complete your revisions, please reply to this message or contact the journal office at plosone@plos.org. Please include the following items when submitting your revised manuscript:

We look forward to receiving your revised manuscript.

Kind regards,

Mohar Singh, Ph.D. Plant Breeding

Academic Editor

PLOS ONE

Journal Requirements:

3.We suggest you thoroughly copyedit your manuscript for language usage, spelling, and grammar. If you do not know anyone who can help you do this, you may wish to consider employing a professional scientific editing service.  

Additional Editor Comments (if provided):

Manuscript need to improve as suggested

Reviewers' comments:

Reviewer's Responses to Questions

**Comments to the Author**

1. Is the manuscript technically sound, and do the data support the conclusions?

Reviewer #1: Yes

Reviewer #2: Partly

2. Has the statistical analysis been performed appropriately and rigorously? 

Reviewer #1: Yes

Reviewer #2: Yes

3. Have the authors made all data underlying the findings in their manuscript fully available?

Reviewer #1: Yes

Reviewer #2: Yes

4. Is the manuscript presented in an intelligible fashion and written in standard English?

Reviewer #1: Yes

Reviewer #2: No

5. Review Comments to the Author

Reviewer #1: Although the authors have done good piece of research work providing the complete insight of changes occurred during the course of domestication of grapevine. But the length of manuscript need to shorten by removing of redundancy and should be presented in concise form.

Reviewer #2: The manuscript presents the effect of pip shape and size affecting size of grape berries. Though relevant but preliminary and also do not add much new to the existing knowledge. Experimentation lacks uniformity e.g., sampling from wild grown and cultural conditions thereby raises questions on varied effect of soil, climate, pruning intensity etc. Only a single year study

is insufficient to draw valid conclusions thus warrants comprehensive investigations. May be considered as a short communication than a full length paper. Language needs to be improved for more clarity and ambiguity be removed.

6. PLOS authors have the option to publish the peer review history of their article (what does this mean?). If published, this will include your full peer review and any attached files.

Reviewer #1: No

Reviewer #2: No

---

## [Author Response · Author response to Decision Letter 0]

31 Jul 2020

Data availability:

>>> We made the datasets used in this study available in figshare. The reserved doi (doi:10.6084/m9.figshare.12696602) has been properly cited in the MS and will be active upon final acceptance. For now on, they can be accessed through this private link: https://figshare.com/s/959e803083001862ef3d

1. Is the manuscript technically sound, and do the data support the conclusions?

Reviewer #1: Yes

Reviewer #2: Partly

>>> We hope that the changes made now make the MS fully meet PLoS One standards.

4. Is the manuscript presented in an intelligible fashion and written in standard English?

Reviewer #1: Yes

Reviewer #2: No

>>> The MS has been written by a non-native (myself) but reviewed before the initial submission, and rereviewed now.

5. Review Comments to the Author

Reviewer #1: Although the authors have done good piece of research work providing the complete insight of changes occurred during the course of domestication of grapevine. But the length of manuscript need to shorten by removing of redundancy and should be presented in concise form.

Reviewer #2: The manuscript presents the effect of pip shape and size affecting size of grape berries. Though relevant but preliminary and also do not add much new to the existing knowledge. Experimentation lacks uniformity e.g., sampling from wild grown and cultural conditions thereby raises questions on varied effect of soil, climate, pruning intensity etc. Only a single year study

is insufficient to draw valid conclusions thus warrants comprehensive investigations. May be considered as a short communication than a full length paper. Language needs to be improved for more clarity and ambiguity be removed.

>>> The two reviewers agree about the length of the paper and asked to shorten it. We followed their advice and removed ~600 words (>12%) of the main MS body and did our best to keep it intelligible yet to make it more concise.

>>> The reviewers disagree about the novelty of the results brought here. The reviewer n¬∞2 stated that they "do not add much new to the existing knowledge". 

We here disentangle many completely new aspects of the complex interplay between grapevine biology, development, and domestication using the pip and the berry forms as entry points. It is absolutely true that some long-known patterns (eg the shape difference between the wild and domesticated types) are not revealed but confirmed here, yet on a larger and more representative dataset. These totally new results presented here were presented as preliminary steps towards testing possible (and long lasted) biases in archaeobotanical inference such as the effect of the number of pips in a berry on their forms.

The reviewer n°2 also states that "sampling from wild grown and cultural conditions thereby raises questions on varied effect of soil, climate, pruning intensity etc.". 

Differences indeed exist and that is exactly what we were looking for. For the analyzed traits, our results indicate that, half of what was supposed to be a "domestication gap" seems to be caused by cultivation. 

We tempered a few statements but we have the feeling that the new questions brought by our findings are presented as promising perspectives, not definitive conclusions. It the Editor has the feeling our Discussion should be tempered once again, we will be happy to do it.

We hope that the changes made improve our manuscript and now fully satisfy both the reviewers and the editors.

Yours sincerely

Vincent Bonhomme

---

## [Decision Letter · Decision Letter 1]

15 Sep 2020

Eco-evo-devo implications and archaeobiological perspectives of trait covariance in fruits of wild and domesticated grapevines

PONE-D-20-10713R1

Dear Sir,

We’re pleased to inform you that your manuscript has been judged scientifically suitable for publication and will be formally accepted for publication once it meets all outstanding technical requirements.

Kind regards,

Mohar Singh, Ph.D. Plant Breeding

Academic Editor

PLOS ONE

Additional Editor Comments (optional):

Accepted

---

## [Editor Report · Acceptance letter]

6 Oct 2020

PONE-D-20-10713R1 

Eco-evo-devo implications and archaeobiological perspectives of trait covariance in fruits of wild and domesticated grapevines 

Dear Dr. Bonhomme:

I'm pleased to inform you that your manuscript has been deemed suitable for publication in PLOS ONE. Congratulations! Your manuscript is now with our production department. 

Kind regards, 

on behalf of

Dr. Mohar Singh 

Academic Editor

PLOS ONE